# Similarities of Minoan and Indus Valley Hydro-Technologies

**S. Khan [1],*, E. Dialynas [2],*, V. K. Kasaraneni [3]**  **and A. N. Angelakis [4]**

[1] Institute of Social Sciences and Directorate of Distance Education, Bahauddin Zakariya University, Multan, Punjab 60000, Pakistan

[2] DIALYNAS SA, Environmental Technology, Troulos Kallitheas, GR 71601 Heraklion Crete, Greece

[3] Department of Environmental Sciences and Engineering, Gannon University, Erie, PA 16541, USA; kasarane001@gannon.edu

[4] HAO-Demeter, Agricultural Research Institution of Crete, 71300 Iraklion and Union of Water Supply and Sewerage Enterprises, 41222 Larissa, Greece; angelak@edeya.gr

\* Correspondence: saifullahkhan33@gmail.com (S.K.); md@dialynas.com (E.D.)

**Abstract:** This review evaluates Minoan and Indus Valley hydro-technologies in southeastern Greece and Indus Valley Pakistan, respectively. The Minoan civilization first inhabited Crete and several Aegean islands shortly after the Late Neolithic times and flourished during the Bronze Age (*ca* 3200–1100 BC). At that time, the Minoan civilization developed fundamental technologies and reached its pinnacle as the first and most important European culture. Concurrently, the Indus Valley civilization populated the eastern bank of the Indus River, its tributaries in Pakistan, and the Ganges plains in India and Nadia (Bangladesh), spreading over an area of about one million km$^2$. Its total population was unknown; however, an estimated 43,000 people resided at Harappa. The urban hydro-technologies, characteristics of a civilization can be determined by two specific aspects, the natural and the social environment. These two aspects cover a variety of factors, such as climate and social conditions, type of terrain, water supply, agriculture, water logging, sanitation and sewerage, hygienic conditions of communities, and racial features of the population. Therefore, these factors were used to understand the water resources management practices in early civilizations (e.g., Minoan and Indus Valley) and similarities, despite the large geographic distance between places of origin. Also discussed are the basic principles and characteristics of water management sustainability in both civilizations and a comparison of basic water supply and sanitation practices through the long history of the two civilizations. Finally, sustainability issues and lessons learned are considered.

**Keywords:** Ancient Greece; Ancient Pakistan; Hydrology; planning; storm-water harvesting; water management; water supply

## 1. Prolegomena

Minoan, a great civilization of the Bronze Age (*ca* 3200–1100 BC) was established on the island of Crete, in the Aegean islands and western coastal areas of today's Turkey. The people of Minoan lived in harmony with the environment unlike any other European civilizations of the era. Crete, located in the eastern Mediterranean, was the center of Minoan culture. Crete is the largest island of Greece, covering an area of 8336 km$^2$ with about 1000 km of coastal line [1].

The Minoan Era was called by Arthur Evans [2] the *Pax Minoica* or "Minoan peace", a time without any conflict or/and war. Very little is known about the origin of the first immigrants in prehistoric Crete, or of the principal genetic relationships between the Minoans, Neolithic, and ancient civilizations and

modern European populations [3]. Studies reported the possibility that the Neolithic settlers in Crete originated from the Middle East and Anatolia. At the height of its greatness, Crete had 90 populous cities (Homer, Odyssey xix), including Knossos, with an estimated 100,000 people in 1600 BC [4,5]. Minoan Crete reached its peak population density during the late Minoan period (after *ca* 1450). These estimates are based on numerous known palaces, cities, villages, and other Minoan settlements.

The basic skills and characteristics of Minoan architecture can be seen clearly in the Minoan palaces (Knossos, Malia, Phaestos, and Zakros) on the island of Crete [6] (Figure 1). One of the salient features was the architectural and hydraulic operation of its water supply, sewerage, and drainage systems in Minoan palaces, cities, and other settlements [2]. It was followed by a long period (*ca* 1150–800 BC), usually referred to as the "Dark Ages" due to the regression in technological advancement before the advent of the Archaic, Classical, and Hellenistic periods. The hydro-technologies developed by Minoans were further improved during these periods.

Another ancient civilization that flourished around the same time period is the Indus Valley civilization. Historians and archaeologists have not completely deciphered the writing system—the data come mainly from archaeological excavations. It is believed that the Indus Valley civilization, at its pinnacle, affected an area much larger than Mesopotamian (e.g., Sumerian and Assyrian) and Minoan civilizations, with more than 1500 cities at its peak [7,8]. Geographically, it covered an area of about one million square kilometers. The most notable sites are Moen-Jo-Daro, Harappa, Kot Diji, Dholavira, Rakhigarhi, Kalibangan, and Lothal (Figure 1) [9].

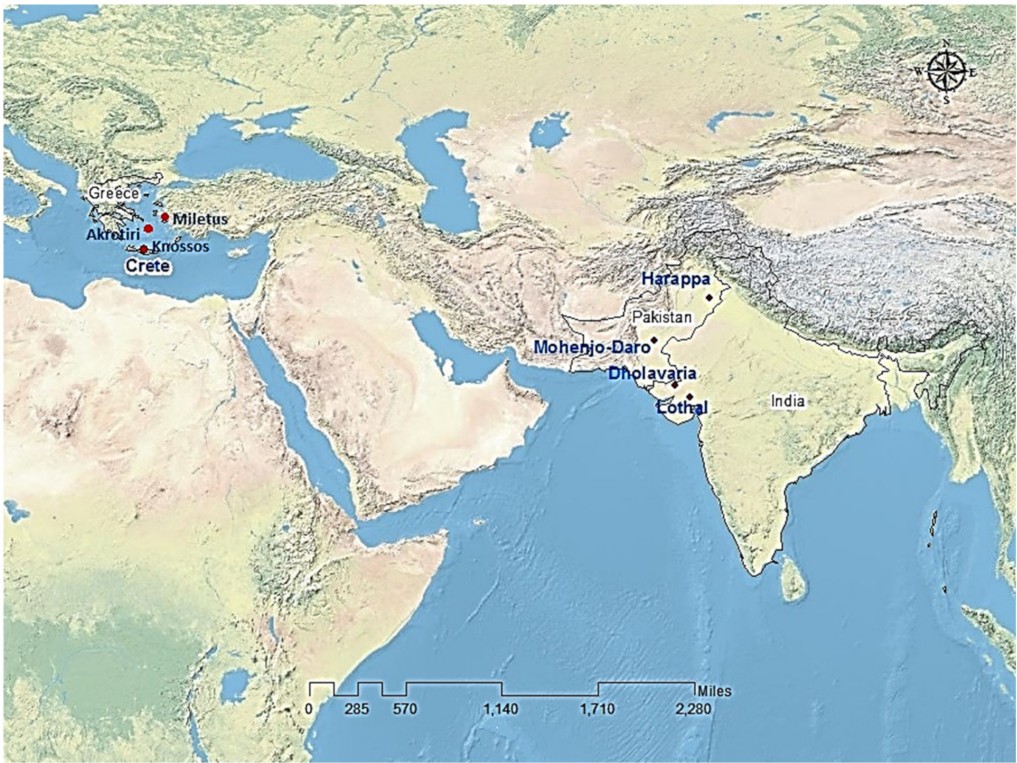

**Figure 1.** Map of areas considered: Minoan and Indus Valley Civilization.

The origins and the time of the first human settlers in the Indus Valley region are unknown. It was hypothesized that perhaps the first people arrived by sea from Africa and settled southwest of the Indus Valley. Northern migrants (Gandhara civilization) may have arrived through the Khyber Pass in the Hindu Kush Mountains. Archaeological evidence found in the highlands of agricultural lands dating to about 7000 BC revealed that people were farming in several places along the Indus River since *ca* 3200 BC [10].

Indus Valley citizens were well-known for their urban planning. Indus Valley cities were well-defined grid systems. Early hydro-engineers also formed functional plumbing and sewage systems. These systems were comparable to any urban drainage and sewerage systems established later in the 19th century. The consistency in the urban planning and residential construction indicates that the Indus people had developed a strong central and local government system [11]. It should be noted that several of the Greek cities later in the Classical period were built on a similar plan, known as the Hippodameian plan, named after Hippodamos of Miletus (498−408 BC). All the streets were parallel, forming a grid, and from 5 to 10 m wide [12]. In addition, sewers were constructed under the pavements. According to Aristotle, Hippodamos was the first planning architect to design a city in the form of a grid.

In this review, the urban hydro-technologies of Minoan and Indus Valley civilizations, with reference to similarities, are presented and discussed. The specific objectives are the study of historical hydro-technologies, to assess their similarities, and to analyze their development during the Bronze Age. The research tools used for the study include field visits, literature review, observations, discussions, correlation, and presentations available on cyber media, as well as different archeological museums.

This review presents evidence and discusses the similarities between both civilizations in the following order: (a) summary of urban and regional planning; (b) water and sanitation technologies; and (c) the similarities of hydrotechnologies among the two civilizations. Finally, the Epilogue: lessons learned and concluding remarks are included.

## 2. Urban and Regional Planning

### 2.1. Urban Centers

The Minoans began building palaces at the start of the Protopalatial period (*ca* 1900 BC) to act as cultural, commercial, religious, and administrative centers for their society, which was rapidly expanding at the time. The palaces were built on hills in strategic locations in a manner so complex that they appeared like labyrinths from outside [13]. Each palace was built in an impressive way on the same basic layout, which indicates a strong system of organization. The primary construction materials used were untreated stones and ceramic bricks bound by mortar. Minoan palaces excavated to date were not surrounded by defensive walls, and neither were most palaces built by subsequent Greek civilizations. The palaces were multi-storied buildings and they included both exterior and interior stairs, light wells, huge columns, storage rooms, open-air courtyards, and the forerunner of the ancient theaters [14]. The cities were technologically advanced and well operated, with water supply and sewerage and drainage systems that provided good sanitation conditions for the inhabitants.

City planning was one of the most outstanding features of the ancient Indus civilization, which reveals that the civic organization of cities was highly developed. Similar to the Minoans, the Indus cities had parallel streets forming a grid system dictated by functional reasons, separate living apartments, flat-roofed houses constructed of bricks, and efficient water supply and drainage and sewerage systems with appropriate ventilation [15]. The cities included streets, houses, large buildings, industrial sites, and dockyards that were very well planned and constructed. The administration system was a centralized type. The primary construction materials were bricks, while stones and wood were also used in the construction of buildings [16]. The houses were built on plinths that rose above the street level with stairs located near the front door wall. The public places were planned and constructed differently; dwellings ranged from single to several public rooms, with baths and common wells and sanitation infrastructures. The city sewerage and drainage systems were well designed and appear to have been effectively operated and maintained.

### 2.2. Streets and Pavements

Minoan and Indus valley cities were well connected by a network of roads, which were well drained, and had water and sewage distribution systems. Paved street networks that connected

the palaces and cities in Minoan Crete were constructed in the mid-Minoan period (*ca* 1750–1450). Well preserved paved streets, usually narrow, appear in the Minoan ruins of Gournia and Palaikastro and other Minoan sites [17]. A good example is the ancient street in Knossos known as Royal Road, which is considered as the oldest street in Europe (Figure 2a). Royal Road connects the [18] theater area in the main palace with the small palace and is very well paved, with stone made slabs and drains on both sides. Several grandstands may have been constructed in the mid-Minoan times and possibly rebuilt at the very end of the late Minoan times [19].

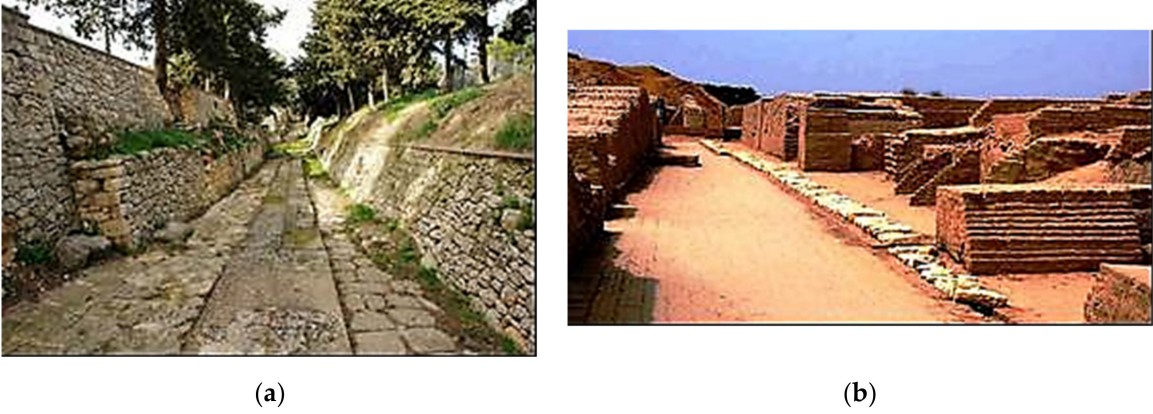

(**a**)　　　　　　　　　　　　　　　　　　　　　(**b**)

**Figure 2.** Streets with paved roads including rainwater drains: (**a**) Minoan (Photo A. N. Angelakis) and (**b**) Moen-Jo-Daro [16].

The Indus Valley civilization also thrived, with well-planned cities and elegant streets laid out in a grid pattern. The longest street in Moen-Jo-Daro was 805 m long and about 10 m wide, which likely indicates that wheeled carts were in existence during this period and streets were used for trade [20]. The city center was connected by main streets from four directions to reduce congestion and allow free movement of people and goods to the central business district. One outstanding feature is a concentric shaped street in the form of a ring structure which was used to control expansion of the cities (Figure 2b). The main streets of Moen-Jo-Daro and Harappa were layered from east to west and north to south, intersecting one another at right angles with width varying from 3.50 to 10.00 m and up to 1.5 km long, that served as flood protection. The lanes were joined with the streets. Each lane had a public use and was provided with street lamps. The life in the Indus cities implies a democratic urban economy like that of the Minoans.

## 3. Water Technologies

In this section, the major hydro-technologies relevant to water harvesting and conservation and water supply and distribution systems (e.g., Aqueducts, cisterns, tanks, dams, and wells) are considered.

### 3.1. Aqueducts

Long-distance systems for transporting water to the urban areas had been developed since prehistoric times due to the mountainous terrain of Crete [21]. Conduits carried water from mountain springs to Minoan palaces and cities, utilizing a combination of open channels and closed pipes as in Knossos [22,23]. Although the Minoan people of Knossos depended partially on wells, they primarily relied on water provided by the Kairatos River, located on the east side of the low hill on which it was built. It has also been suggested by A. Evans [2] that the water supply system for the palace tapped water from the *Mavrokolybos* spring [2]. As population increased, other water sources were developed from further distances and incorporated into the water supply system. A conduit made of terracotta was used to convey water from a water spring on the Gypsadhes hill [24] to Knossos and was later extended to carry water from Mt. of Juctas, *ca* 10 km away [6].

The people of the ancient Indus Valley civilization were pioneers in hydraulic works and had developed and successfully used water management devices. In Dholavira, Lothal, Harappa, and Moen-Jo-Daro, aqueducts were used to convey water from the reservoirs, wells, and water tanks to the city center. Also, water was transported to the agricultural lands for irrigation using canals. The ancient Indus people were experts in building aqueducts as well as drains and sewers. At most Indus Valley sites, sewerage and drainage systems were found to be in good condition. At Harappa, Moen-Jo-Daro, and Lothal, there were open or covered systems built below the street level. The appropriate usage of gradient for drains and/or sewers to assure proper water flow indicates that the people of ancient Indus had very good knowledge of hydraulic engineering. In addition, the carrying capacity was regulated in an appropriate manner [25]. The irrigation works were implemented with smaller scale channels by: (a) diverting river water into them or (b) lifting water from wells using the shaduf system [26].

### 3.2. Cisterns, Reservoirs, and Rainwater Harvesting

Water cisterns usage was a very well-known practice by the Minoans. Technical characteristics and records of Minoan cisterns are reported by Angelakis (2016) [27]. Essentially, the system had been extensively employed by almost all later civilizations in Ancient Crete. Cisterns were used to store both rainwater and spring water, which was conveyed to the urban areas by aqueducts [6,28]. The reservoirs were typically circular structures, assembled with rocks beneath the earth exterior, having diameters of 1.5 m to 7.0 m and depths of 2.5 m to 5.0 m. An earlier Minoan water storage system was found at the focal point of a dwelling compound with origins during the third to second millennium BC in Chamaizi [29]. Similar expertise was employed in the construction of Zakros and Phaestos fortresses (Figure 3a). The reservoirs were joined by means of diminutive channels, which were designed to amass run-off from the highlands and plain areas [30].

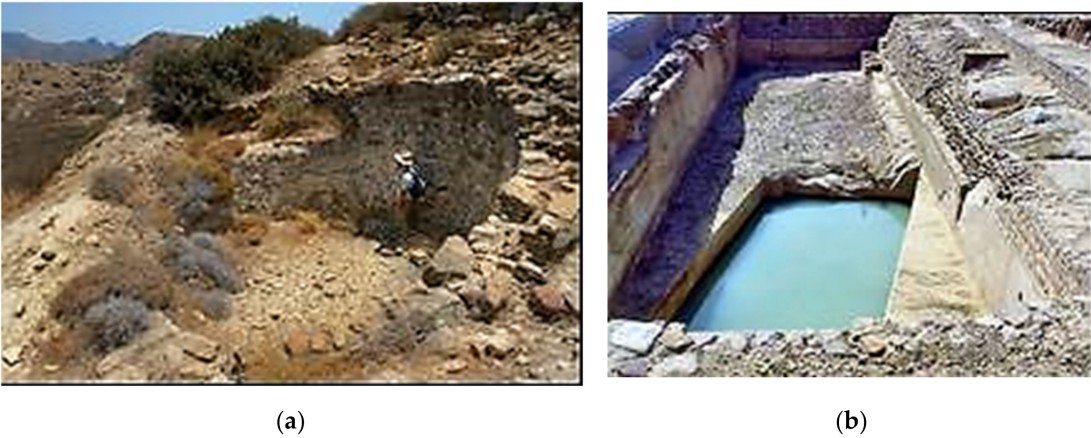

(**a**)                                                                 (**b**)

**Figure 3.** Rainwater harvesting: (**a**) Minoan circular cistern (Photo A. N. Angelakis) and (**b**) reservoirs at Lothal [31].

The Indus Valley civilization developed a very sophisticated rainwater harvesting system. According to Subramanian (2010), in Dholavira and Lothal (India), a series of water reservoir were used for water supply, crop cultivation, and water logging systems [32]. The most notable characteristics of the Dholavira hydraulic system was the sophisticated rainwater storage system consisting of conduits and basins. It is one of the most primitive systems in the ancient world and was entirely made of boulders. To date, three such systems have been found in excavations. One of the dry channels that flowed in north-south direction of the reservoir was logged at many places for harvesting rain water (Figure 3b). The great bath at Moen-Jo-Daro is also evidence of the water conservation and storage system [33]. The residents of Dholavira constructed more than 16 reservoirs with differing shapes and sizes. Some of the reservoirs were constructed to take advantage of the altitude of the land within the mega cities, using a slope of 13 m from northeast to northwest direction [34].

The awe-inspiring scene at Lothal is a monstrous dockyard spreading over a zone of 37 m west to east and about 22 m north to south [35]. Some archeologists are convinced that the structure was utilized as a harbor, but a few believe this structure was an extensive water tank or water reservoir.

Archaeological exploration revealed that the people in the mountainous Sindh province of Pakistan had understood the use of rainwater and launched an outstanding water logging system called Gabar Bund, in which rainwater was diverted for irrigation as well as domestic uses. The Gabar Bund system in the Baluchistan province (Mehrgarh) originated in the Chalcolithic Nal period (*ca* 6500). The word "bund" means a retaining wall-like structure and the expression "Gabar" has been linked by historians to the "Medieval Iranian Zartushtis period (*ca* 750–550), which is in addition known as Parsees or Persians" [36].

### 3.3. Dams

Dams have been built in Mehrgarh and Mesopotamia since the Neolithic times, *ca* 7000–3200 BC. Thereafter, throughout the Bronze Age (*ca* 3200–1100 BC), dams were built in southeastern Greece and the Indus Valley to make the cities more adaptive to flood hazards, and to improve the living standards of the people [37]. Minoan dams have been discovered in Pseira, Gournia, and Choiromandres [37]. In order to ensure water supply for domestic use, irrigation, and livestock, the Minoans built dams for water management purposes. At several Minoan sites, the water management systems were based on: (a) small dams to intercept runoff waters in the seasonal streams, (b) larger dams for collecting and storing water, and (c) dams for diverting surface water, mainly from rivers, which in Minoan flowed year round [37,38]. The dams were principally constructed of stones and clay mortar [39]. A water diversion dam at the river, which bordered the city of Gourna on the west side, has been described by Baba *et al.* (2018) [37]. Over the river bed, around 120 m from the present shoreline, the structure comprised of two external dividers constructed from gigantic cyclopean stones with an aggregate width of roughly 4 m [40], which was used for water supply to the city.

The Indus Valley civilization settled along the Indus River and built up broad underground urban pipe systems, including channels and water funnels for conveying water to the mega cities of Harrapa and Moen-Jo-Daro. A similar hydro-structure to that used by Minoans was also used by Indus people to divert surface flow water into reservoirs located at the center of the cities [41].

### 3.4. Wells

The water supply in the southeastern Minoan Crete was dependent on groundwater. In the town of Palaikastro, Knossos, and palace of Zakros [6], several wells were found, with depths varying from 10 to 15 m [28]. A few wells were located at the northern part of the Minoan Amnissos, a settlement on the north shore of Crete, used as a port in the palace of Knossos. These wells, designated by the name "Villa of the Lilies", were most likely dug in the early late era and reconstructed with the use of ashlar blocks in later years [42].

The people of Moen-Jo-Daro had expertise in digging water wells. The residents of the city had built about 700 wells, where a single well serviced three houses [43]. The wells were assembled with tightening mud bricks to have a stronger life to last for centuries [44]. Scanty winter rains swelled the populations of the cities from the surrounding countryside. Wells were constructed far from the Indus river and its tributaries, unlike other major cities of the Indus Valley civilization. Therefore, the administrations of the urban centers must have felt the need to have more wells for domestic use and log the extra flow of the river and rain water for different domestic and cultivation purposes.

### 3.5. Water Distribution and Fountains

Both the Minoan and Indus Valley civilizations used terracotta pipes for water distribution systems in entire cities. In the Knossos and Malia palaces and in cities such as Tylissos, the terracotta piping for the water supply was placed under the floors at depths up to 3 m [6,45].

In addition to palaces, the Minoan and Indus Valley civilizations also had baths set-up for foot-washing, bathtubs and showers furnished with hot-water and spring-chambers [2]. Fountains are known to have existed since the early Bronze Age in southeastern Greece and the Indus Valley. A typical Minoan fountain, known as a *Tykte*, was located in the west of the Caravanserai, containing a spring-chamber with a wash basin assumed to have to played the role of waterlogged tub or reservoir [46]. An additional *Tykte* fountain was excavated in the Zakros palace of Minoan with dimensions of about 3 m × 4 m (12 m$^2$) and dated back to the late Minoan era [47].

The people of the Indus Valley civilization also constructed underground and surface water tanks and used terracotta pipes to supply water into the houses. These water tanks were constructed adjacent to a main well and were likely used to lift water from the main wells and fill nearby tanks for storage and supply to the household using shadufs [48]. Shaduf is a hand-operated device with a long, tapering, nearly horizontal pole mounted like a seesaw for lifting water. Terracotta pipes used for washing basins and manholes constructed by the Indus Valley people are still in good condition after nearly five thousand years.

In Indus Valley civilization, every residence had water channels with a fountain for drinking water and bathing. Terracotta pipes connected the sewerage system under the nearby street. Furthermore, water from fountains was carried off to the sewage pipes in the streets. It is expected that this technology was a new idea of the Indus and Minoan civilizations.

## 4. Sanitation Technologies

The specific functions of sewerage and sanitation skills promoted by these two civilizations are discussed in the subsequent sections.

### 4.1. Sewerage and Drainage Systems

The Minoan civilization had the capability to establish drainage and sewerage systems in several palaces, which are functioning properly to date. As the hill in the palace of Knossos occasionally received large amounts of precipitation, the development of a drainage system was necessary. An open channel sewerage and drainage system of a total length of 150 m was constructed using boulder slabs approximately 79 cm × 38 cm per channel, lined up by means of clay or mud (Figure 4a). The sewers and drains were large enough to allow humans to enter for maintenance and cleaning. For locations where the channels were completely closed for long distances, manholes were constructed for maintenance. The air-shafts at equal distances were provided for the purpose of aerating the sewer systems [24]. Similar drainage and sewerage systems were found in several other Minoan sites, of which the most significant is the excavation in the Hagia Triadha, located on the southern coast of Crete (Figure 4a) [30].

This system is admired by several visitors to date. The Italian writer A. Mosso, who visited Phaistos and Hagia Triadha during a heavy rain, noticed that the sewers/drains functioned perfectly, and he recorded the incident, saying: "I doubt if there is other case of stormwater drainage system that works 4000 years after its construction" [39]. Also, the American H. F. Gray (1940) reported that: "You can enable us to doubt whether the modern sewerage and drainage systems will operate at even a thousand years" [26].

In the past few decades, archaeological explorations carried out at Harappa, Moen-Jo-Daro, and Lothal revealed a number of incredible observations. Town planning in Moen-Jo-Daro, Harappan, and Lothal, in addition to palaces, had numerous homes, from double-roomed to multi-roomed tall buildings [17]. The town of Moen-Jo-Daro had a more reliable drainage, water supply, and sewerage system which was constructed with mud bricks, which were highly functional and more reliable compared to modern times. The granaries were also wisely assembled, with planned platforms and air ducts (Figure 4b).

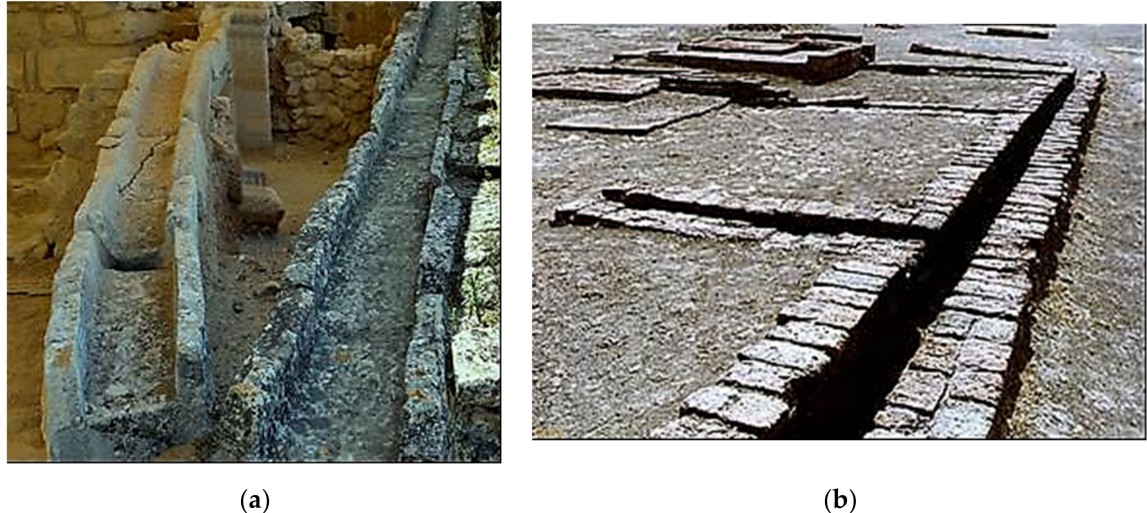

(**a**) (**b**)

**Figure 4.** Sewerage and drainage systems: (**a**) Minoan (photo A. N. Angelakis) and (**b**) Indus valley [31].

In the Indus Valley civilization, the drainage systems of the residences were connected with sewers linked into the main public conduits. The sophisticated drainage pattern and construction of the Harappa residences indicates that they developed a high sense of healthiness and hygiene (Figure 4b). The drainage system and drains were covered with bricks or stones, with inspection traps and manholes at regular intervals. Every house had their own soak pit, which collected all the residual solids and allowed only the water to surge into the main sewer drains [49]. Finally, the sustainability principles of those applied technologies is remarkable.

*4.2. Bathrooms and Lustral Basins*

Even though it is difficult to define the characteristics of Minoan rooms, three different rooms were identified as bathrooms during the excavation of the Knossos palace. One style resembles a bathroom excavated at Malia (Figure 5a), Phaestos, and Knossos. However, there is a variation in the construction level of the base and the subsequent dearth of the steps of the bathroom. Evans [2] has referred to these rooms as "Lustral chambers or basins". Moreover, Walter (1984) presumed that Lustral chambers or basins were in the Queen's palace; later, however, turned into public bathrooms. Further, the investigations by Platon (1990) [50] led to the assumption that the Queen's bathroom was converted to a public bath in the palaces at Tylissos and Knossos [24]. Consequently, it was observed that in the conclusion of the Minoan civilization, decay commenced in hygienic conditions prior to spiritual ceremonies like godliness.

In general, the ruins of Moen-Jo-Daro show a picture of a community in which both personal and community cleanliness was prioritized and practiced. Almost all houses in Moen-Jo-Daro had a bathroom, always constructed at the side of the building adjacent to the street for the convenient disposal of wastewater into the street drains [51]. Latrines found in the houses were placed close to the street wall for the same reason. Ablution places were placed adjacent to the latrines, thus conforming to one of the most modern of sanitary maxims. Bathrooms and latrines situated on the second floor were drained using terracotta pipes with a fitting valve placed in the boundary wall of the building (Figure 5b).

In some houses in Harappa and Moen-Jo-Daro, it was observed that the people might have stood on bricks or shower tray in the bathroom and poured water on themselves manually using a mug. The fresh water was supplied from an underground well adjacent to the houses and the sewer water flowed through a terracotta pipe to a nearby channel in the lane [52]. The "Great bath" in front of the King's Priest House in Moen-Jo-Daro has undoubtedly the most primitive public swimming pool or

water tank of the Ancient civilizations. The swimming pool was roughly 12 m in length from north to south and 7 m wide from east to west (Figure 4b), with a depth of 2.4 m [26].

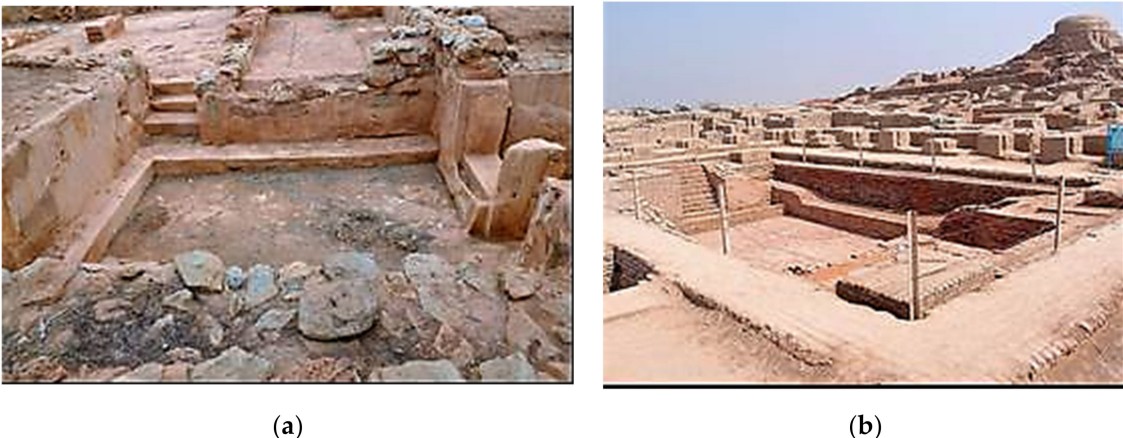

| (a) | (b) |

**Figure 5.** (**a**) Bath in the house of Da at the palace of Malia (photo A. N. Angelakis) and (**b**) King swimming pool at Moen-Jo-Daro [31].

### 4.3. Toilets or Lavatories

In general, with time, the public administration of any society will be obligated to provide toilet facilities to low-income residents. In Minoan and Indus Valley civilizations, there was a long history of communal lavatories and these toilets were constructed and controlled by the municipalities [53].

Sewers, bathrooms, toilets, and other sanitation structures were considered necessary features in most Minoan palaces. A room of interest was identified by Evans [2] in the palace of Knossos as a toilet with a wooden seat (Figure 6a). As with modern toilets, it had a flushing system with a conduit under the seat leading the outflow from the toilet to the outside sewer [5]. Similar toilets known as "Da" were excavated in a house in the vicinity of the palace at Malia [54].

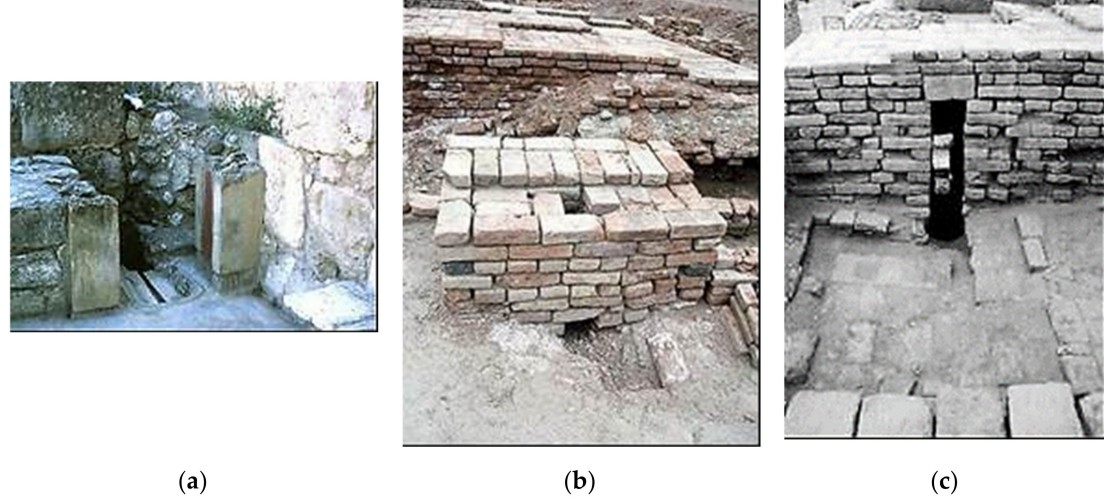

| (a) | (b) | (c) |

**Figure 6.** Toilets: (**a**) at palace of Minos (Knossos) (photo A. N. Angelakis) and (**b**,**c**) brick toilets at Harappa and Moen-Jo-Daro cities [31].

In *ca* 2600 BC, the affluent classes in Moen-Jo-Daro constructed public toilets. In rural areas, the majority of the civilians covered older houses and utilized open cavities in the ground as toilets (Figure 6b,c). The inhabitants of the Indus Valley civilization had prehistoric water-cleaning toilets

with running water in all quarters and had sewer lines connected to mains drains plastered with bricks or wood. The running water of the channel from a nearby well was used for the removal of human waste from in the toilets [55]. Amongst other contributions, both civilizations introduced the world's first known "flush toilets", with many houses with washing platforms and supreme toilets or waste disposal holes. The flushed waste fed into a nearby soak pit through brick pipes and channels. The soak pits were regularly emptied of the human waste, which may have been used as fertilizer for agricultural fields, along with the animal wastes [49].

### 4.4. Disposal and Reuse Sites

Land application of sewage for agriculture irrigation and existed since the beginning of the Bronze Age (*ca* 3200–1100 BC) in the Indus Valley and in Crete, Greece [56].

During the Minoan Era, southeastern Greece periodically experienced severe water shortages [57]. Thus, during those periods water reuse was a necessity. In order to maximize and utilize the inadequate water resources available, it was necessary to reuse water. The disposal sites were differentiated among the Minoan settlements. For example, in the palaces of Zakros and Knossos the wastewater and stormwater were directed to the sea and Kairatos river, respectively. On the other hand, in the palace of Phaestos, wastewater collection facilities were used to divert wastewater and stormwater to farmland located southeast of the palace.

Similarly, in the Indus Valley civilization waste disposal of agricultural lands was practiced in the cities of Harappa and Moen-Jo-Daro [58]. The Harappan city had very efficient drainage and sewerage systems. The central sewer/drain was connected to every house, making certain of efficient disposal of wastes. For maintenance, openings were provided along the length of the entire system and on the streets. The sewers/drains were covered and connected to the larger sewerage/drainage outlets, which guaranteed the disposal of waste away from the city [59].

The wastewater from bathrooms and kitchen, the latrines, and roof drainage typically entered the street drains indirectly via tightly brick-lined conduits, with outlets to the street drains. These pits were inspected from time to time for maintenance, being the settling basins or soakage pits located along the street sewers/drains [8]. Houses also had vertical chutes built into the walls and descending from the upper floor. At the base of these ducts oftentimes bins were provided at the street level which could be cleaned out by the scavengers. Public rubbish bins were also provided at convenient places [26].

## 5. Discussion

In this brief comparative review of two of the most important prehistoric civilizations, a number of interesting parallel achievements between the two civilizations and potential cultural interactions as well as the sustainability of their hydro-technologies have been identified.

Both civilizations had very a modern concept of planned layout of their settlements. In the Indus Valley, the entire residential areas were built in a grid pattern based on the benefits of the parallel streets, which were dictated by functional reasons. The engineers who planned these layouts had very good knowledge of the optimum utilization of space. In the Minoan civilization the palaces were built in an impressive way on the same basic layout, with a large, open court in 2:1 length and width oriented on a north-south axis, indicating a strong system of organization. In both civilizations, the central or side walls in different palaces, as well as middle class houses, were intermittent, having several openings and colonnades for free movement of light and air, contact with the outside world, neighbors, and/or to use it as a safe route in case of emergency or war. They had patios, external adjacent rooms, courtyards, and numerous light wells.

Historical and archaeological evidence indicates that advanced water supply and sanitation technologies were practiced in Minoan Greece and the Indus Valley during the Bronze Age. It is known that during the middle Minoan period (*ca* 2000–1750 BC) and especially the beginning of the late Minoan period (*ca* 1750–1100 BC) a "cultural explosion" occurred in Crete and several Aegean islands. This is assured, inter alia, by the advanced water management techniques which were practiced at

that time and had a high degree of similarity to those found in the Indus Valley urban areas from *ca* 5000 ago. These systems, such as those in the cities of Harappa and Moen-Jo-Daro, made a thriving civilization possible. In the city of Harappa, every house was connected to the central sewerage and drainage system, ensuring the proper removal of wastewater. Inspection manholes were provided in order to insure that the systems operated and were maintained properly. We do not know why and how that happened, but the fact remains that it happened.

In conclusion, there are many similarities and valuable lessons to learn about water resources management, especially those in urban areas, from both civilizations. These could be summarized as follows:

(a)  Minoans built aqueducts (e.g., Knossos, Tylissos, and Malia) to convey freshwater from the springs to the palaces and other settlements. Water supply was distributed throughout the palace with networks constructed of terracotta pipes with a conical geometry located beneath the palace floors. Similarly, in Dholavira, aqueducts carried the waters from the reservoirs to the city and also to irrigated fields by canals;

(b)  In areas where spring and/or surface water was not available the Minoans and the Indus people built numerous wells (e.g., eastern Crete and Santorin), as well as cisterns in southeastern Greece and huge reservoirs (large scale cisterns) in the Indus Valley, which were filled by rainwater using harvesting techniques similar to those of the Minoans. However, aside from the scale of those technologies, the basic principles of both technologies were identical;

(c)  Dams were constructed by Minoan (e.g., Choiromandres and Pseira in eastern Crete) for regulating the flow of the stream through a water management system of small dams. The small dam system was designed to protect arable land from erosion after heavy rainfall, while a larger dam was constructed for storing runoff from fields. They also built dams for diverting river water (e.g., Gournia). In addition, a very innovative technology (Gabar bund and shaduf) was developed by the Indus people in order to divert the river flow to fill 16 reservoirs that were located in the periphery of the city of Dholavira [60]. These technologies were also utilized at Moen-Jo-Daro city. There are indications that the Minoans had also been using the shaduf system for lifting water since the Meso-Minoan period (*ca* 2100–1600 BC) in Zakros and Palaikastro wells [48];

(d)  Both Minoan and Indus people used the network of terracotta piping located beneath the floors for distributing the potable water;

(e)  The Minoan palaces, as well as the cities in the Indus Valley, had baths and toilets, even on the upper floors, from where the wastewater was carried out by terracotta pipes to the central sewerage and drainage system;

(f)  Both civilizations built drains and sewers for carrying storm- and wastewaters away from the palaces and the cities. In the Indus cities, the drainage and sewerage systems were built using stones and bricks, while in the Minoan palaces stones lined with cement-lined limestone. The sewers/drains were equipped with manholes for the purpose of maintenance;

(g)  Minoans in the later period began practicing cleanliness before ritual events. Thus, bathrooms were considered necessary. Similar in principle, bath-structures were also used by the Indus Valley, but at a higher scale;

(h)  Design and construction principles. Naturally, it is difficult to estimate the design and construction principles of Minoan and Indus Valley "engineers", but it is notable that several ancient works have operated for very long periods, some until recent times. For example, waste- and storm-water drainage systems were functioning for millennia [39];

(i)  The Minoans seemed to be more democratic in their social organization, living in harmony with the environment, whereas the Indus Valley people seemed to be more socialist and their organization was based on the community perception.

From all these similarities in water management and hydro-technologies, the next question is: how did the Minoans gain all the required knowledge for these sophisticated water management practices at the beginning of the Neopalatial period (*ca* 1750–1450 BC), when none of their neighboring civilizations in the area possessed them? Misra (2017) stated that all the water management techniques employed by the Minoans during the Neopalatial period had been in use in the Indus Valley civilization since *ca* 3000 BC. It could be hypothesized that some people living in these sites were aware of the principles in water and storm- and wastewater management. This suggests the existence of master hydraulic technicians responsible for constructing, operating, and maintaining the water supply and sewerage and drainage systems. They could also have been in charge of solving water management related problems. Their job description may have included providing palaces and settlements with cost-effective, environmentally friendly potable water supply and sewage systems. A lot can be learned from Minoan and Indus Valley technologies and practices. The concept of personal and environmental cleanliness was part of the way in which the Minoan and Indus people organized their societies. Both civilizations were pioneers in technologies which were not utilized by other civilizations for several centuries.

As previously discussed, classical architect and planner Hippodamos of Miletus was initially active in Ionia, and especially in his place of origin, Miletus. He developed the urban plan known as the Hippodameian system, similar to that used by Indus Valley people. The existence of a system strikingly similar to the Hippodameian system in the Indus Valley suggests ancient Greeks were in contact with the ancient Indus people. Recently, a Bronze Age wall painting was identified on the island of Akrotiri (Thera) which depicts a monkey whose native habitat is thousands of kilometers away in Asia [61]. The finding suggests that ancient cultures separated by great distances were trading and exchanging products, ideas, and technologies [62].

## 6. Epilogue: Lessons Learned

The landmark technological achievements by both civilizations had no precedent in prior civilizations nor historical and pre-Byzantine civilizations. The hydro-technical parallels between the two civilizations strongly suggest the two civilizations may have indeed had significant contact with each other [63]. This synopsis of advanced ancient sanitation systems, including their long durability and sustainability, should be concluded with Herold Gray's (1940) [26] statement:

> We frequently hear people speak of modern hygiene as if it something rather recently developed, and there appears to be a prevalent idea that municipal sewerage is a very modern thing that began some time about the middle of the last (19th) century. Perhaps these ideas do something to support a somewhat wobbly pride of the modern civilization [ . . . ], but when examined in the light of history that is far from new or recent. Indeed, in the light of history, it is surprising, if not bitterness, the fact that man has gone so poorly, if at all, in about 4000 years [ . . . ]. Archaeologists researches this [Minoan and Indus] space give us the image that people have come a long way towards a comfortable and hygienic living, with a considerable degree of beauty and luxury [...]. And this was about 4000 years ago [26]

In addition to sustainability aspects, the examples on hydro-technologies presented in this paper are even important for today's water engineering. Some lessons learned include:

(a) Both these civilizations understood the importance of sanitation, water supply, and drainage and sewerage systems for human survival and well-being and made these an essential part of urban planning to achieve water resource sustainability;

(b) Indus Valley and Minoan water technologies were characterized by simplicity, ease of operation, and the requirement of no complex controls, making them more sustainable [47]. Nevertheless, the successful design and operation of some of these systems were massive achievements in engineering;

(c)    A combination and balance of smaller scale measures (such as cisterns for water harvesting systems) and the large-scale water supply projects (such as reservoirs for storage of aqueduct flows) were used by many ancient civilizations thereafter;

(d)    Several hydro-technologies developed by both civilizations should be considered not as historical artifacts, but as potential models for sustainable water technologies for the present and the future;

(e)    The Indus Valley and Minoan people considered water quality and security as one of the critical aspects of the design and construction of their water supply systems. Water security is also a contemporary concern around the world, particularly from the viewpoint of adequate water supply, and more recently, from the viewpoint of safety;

(f)    What we can learn from the ancients, since the prehistoric times, using traditional knowledge, could be a significant factor in solving our water needs, especially for developing parts of the world [64];

(g)    Finally, it should be noted that: (i) over two billion people are living in regions experiencing high water stress today and the number is expected to increase in the future; (ii) over one billion people do not have access to clean and safe drinking water; and (iii) five to ten million people die each year from water-related diseases or inadequacies [65]. There is therefore a vast need for sustainable and cost-effective water supply and sanitation facilities, particularly in cities of the developing world [66]. Applicability of selected ancient water supply management systems (e.g., storage of rainfall runoff facilities) for the contemporary developing world should be seriously considered.

**Author Contributions:** S.K. had contributed to Indus Valley civilization; E.D. reviewed and submitted the manuscript; V.K.K. contributed mainly to Indus Valley civilization and reviewed and editing the manuscript; A.N.A. had the original idea, contributed to the project idea development mainly to the Minoan civilization parts, revised and edited the manuscript. All authors have read and agreed to the published version of the manuscript.

**Funding:** This research received no external funding.

**Acknowledgments:** We greatly acknowledge the in depth review of Larry W. Mays, School of Sustainable Engineering and the Built Environment, Arizona State University, Tempe, AZ 85287-5306, USA, for his significant contribution in improving this paper. For cross-checking the text and constructive comments offered by Kostas Voudouris, Lab. of Engineering Geology and Hydrogeology, Aristotle University of Thessaloniki, Greece, are gratefully acknowledged. We greatly acknowledge M.P. Rao contribution for English editing.

**Conflicts of Interest:** The authors declare no conflict of interest.

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
