# Peer review of "Similarities of Minoan and Indus Valley Hydro-Technologies"

_sustainability, doi:10.3390/su12124897_

Round 1
Reviewer 1 Report
The text has many underlines, texts in italics, which is incomprehensible. The quality of figures 2, 3, 4, 5, 6 is poor (misaligned, of different sizes, unnecessary borders).
Author Response
Point 1: The text has many underlines, texts in italics, which is incomprehensible.
Response: Done. Only those copied from old publications are left
Point 2: The quality of figures 2, 3, 4, 5, 6 is poor (misaligned, of different sizes, unnecessary borders).
Response: Figures are highly improved and will included in the final version before its publication
Reviewer 2 Report
The entire article has a major descriptive characteristic, that is not quite suitable for the level of this journal.
First of all, I must recommend to consider a new perspective of the organization of the structure, content and names of subchapters and paragraphs in order to highlight the general intention of the research.
(eg/ are to scholastic lines 86-87)
Secondly, very important from my point, it is necessary to clarify the objectives, outcomes and methodology of research; in this way the paper will have an increased sound for readers and specialists.
(Figure 1 is not necessary to be inserted in text because has a poor scientific relevance.)
Thirdly, it will be beneficially for the novelty of the article to have a section dedicated to some lessons learned in the field of contemporary hydro-technologies, based on presented historical similarities evidence.
Author Response
Point 1: The entire article has a major descriptive characteristic that is not quite suitable for the level of this journal.
Response: We reviewed it drastically. A lot of text is rewritten and highly improved.
Point 2: First of all, I must recommend to consider a new perspective of the organization of the structure, content and names of subchapters and paragraphs in order to highlight the general intention of the research.
(eg/ are to scholastic lines 86-87).
Response: (a) Its structure was reorganized by adding two subsections and one new section (as Epilogue: Lessons Learned). (b) The sentence in lines 86-87 was deleted.
Point 3: Secondly, very important from my point, it is necessary to clarify the objectives, outcomes and methodology of research; in this way the paper will have an increased sound for readers and specialists.
(Figure 1 is not necessary to be inserted in text because has a poor scientific relevance).
Response: (a) A paragraph was added in the end of Prolegomena to explain the organization of the work.
(b) Figure 1 shows the location of the two civilizations which can provide the reader a context of the distance between civilizations
(c) The objectives and methodology are discussed at the end of Prolegomena section.
Point 4: Thirdly, it will be beneficially for the novelty of the article to have a section dedicated to some lessons learned in the field of contemporary hydro-technologies, based on presented historical similarities evidence.
Response: Several points were added in section titled Epilogue
Reviewer 3 Report
This article contains very important information and answers the journal's theme.
However, I advise the author to rectify the comments that I presented throughout the article, most of these comments relate to minor corrections, then to update them.

Author Response
Point 1: This is a good article except the English needs serious work. However, I advise the author to rectify the comments that I presented throughout the article, most of these comments relate to minor corrections, then to update them.
Response: The comment made by reviewer are addressed. In addition, the authors revised English.
Round 2
Reviewer 2 Report
Could be usefull to be underlined some suplimentary considerations/limitations of study (in the final part of article).